# Extracting Relationships by Multi-Domain Matching

Yitong Li[1], Michael Murias[2], Samantha Major[3], Geraldine Dawson[3] and David E. Carlson[1,4,5]

[1]Department of Electrical and Computer Engineering, Duke University
[2]Duke Institute for Brain Sciences, Duke University
[3]Departments of Psychiatry and Behavioral Sciences, Duke University
[4]Department of Civil and Environmental Engineering, Duke University
[5]Department of Biostatistics and Bioinformatics, Duke University
{yitong.li,michael.murias,samantha.major,
geraldine.dawson,david.carlson}@duke.edu

## Abstract

In many biological and medical contexts, we construct a large labeled corpus by aggregating many sources to use in target prediction tasks. Unfortunately, many of the sources may be irrelevant to our target task, so ignoring the structure of the dataset is detrimental. This work proposes a novel approach, the Multiple Domain Matching Network (MDMN), to exploit this structure. MDMN embeds all data into a shared feature space while learning which domains share strong statistical relationships. These relationships are often insightful in their own right, and they allow domains to share strength without interference from irrelevant data. This methodology builds on existing distribution-matching approaches by assuming that source domains are varied and outcomes multi-factorial. Therefore, each domain should only match a relevant subset. Theoretical analysis shows that the proposed approach can have a tighter generalization bound than existing multiple-domain adaptation approaches. Empirically, we show that the proposed methodology handles higher numbers of source domains (up to 21 empirically), and provides state-of-the-art performance on image, text, and multi-channel time series classification, including clinical outcome data in an open label trial evaluating a novel treatment for Autism Spectrum Disorder.

## 1 Introduction

Deep learning methods have shown unparalleled performance when trained on vast amounts of diverse labeled training data [21], often collected at great cost. In many contexts, especially medical and biological, it is prohibitively expensive to collect or label the number of observations necessary to train an accurate deep neural network classifier. However, a number of related sources, each with "moderate" data, may already be available, which can be combined to construct a large corpus. Naively using the combined source data is often an ineffective strategy; instead, what is needed is *unsupervised multiple-domain adaptation*. Given labeled data from several source domains (each representing, e.g., one patient in a medical trial, or reviews of one type of product), and unlabeled data from target domains (new patients, or new product categories), we wish to train a classifier that makes accurate predictions about the target domain data at test time.

Recent approaches to multiple-domain adaptation involve learning a mapping from each domain into a common feature space, in which observations from the target and source domains have similar distributions [14, 45, 39, 30]. At test time, a target-domain observation is first mapped into this shared feature space, then classified. However, few of the existing works can model the relationship among different domains, which we note is important for several reasons. First, even though data in different domains share labels, their cause and symptoms may be different. Patients with the same

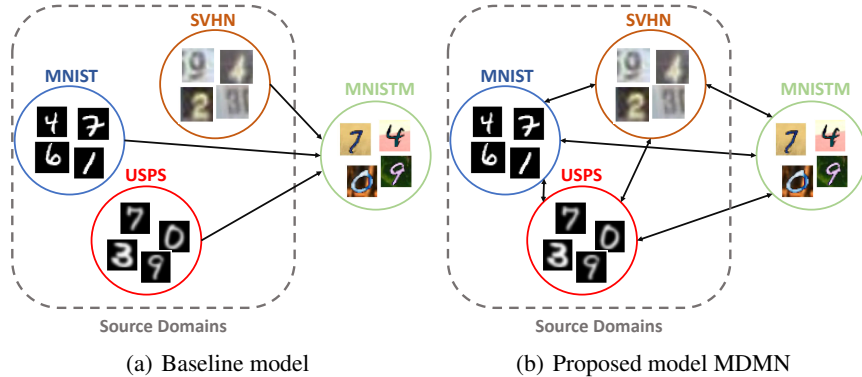

(a) Baseline model          (b) Proposed model MDMN

Figure 1: Figure 1(a) visualizes previous multiple-domain adaptation methods. Figure 1(b) visualizes the proposed method, with domain adaptation between all domains.

condition can be caused by various reasons and diagnosed while sharing only a subset of symptoms. Extracting these relationships between patients is helpful in practice because it limits the model to only relevant information. Second, as mentioned above, a training corpus may be constructed with only a small number of sources within a larger population. For example, we might collect data from many patients with "small" data and domain adaptation is used to generalize to new patients [3]. Therefore, extracting these relationships is of practical importance.

In addition to the practical argument, [32] gives a theoretical proof that adding irrelevant source domains harms performance bounds on multiple-domain adaptation. Therefore, it is necessary to automatically choose a weighting over source domains to utilize only relevant domains. There are only a few works that address such a domain weighting strategy [45]. In this manuscript, we extend the proof techniques of [4, 32] to show that a multiple-domain weighting strategy can have a tighter generalization bound than traditional multiple domain approaches.

Notably, many recently proposed transfer learning strategies are based on minimizing the $\mathcal{H}$-divergence between domains in feature space, which was shown to bound generalization error in domain adaptation [4]. Compared to standard $L1$-divergence, $\mathcal{H}$-divergence limits the hypothesis to a given class, which can be better estimated using finite samples theoretically. The target error bound using $\mathcal{H}$-divergence has the desirable property that it can be estimated by learning a classifier between the source and target domains with finite VC dimension, motivating the Domain Adversarial Neural Network (DANN) [14]. However, neural network usually has large VC dimensions, making the bound using $\mathcal{H}$-divergence loose in practice. In this work, we propose to use a 'Wasserstein-like' metric to define domain similarity in the proofs. 'Wasserstein-like' distance in our work extends the binary output in $\mathcal{H}$-divergence to real probability output.

Our main contribution is our novel approach to *multiple-domain* adaptation. A key idea from prior work is to match *every* source domain's feature-space distribution to that of the target domain [37, 29]. In contrast, we map the distribution ($i$) among sources and target and ($ii$) *within* source domains. It is only necessary and prudent to match one domain to a relevant subset of the others. This makes sense particularly in medical contexts, as nearly all diagnoses address a multi-factorial disease. The Wasserstein distance is chosen to

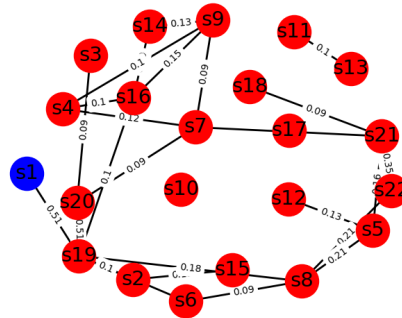

Figure 2: Figure 2 is a visualization of the graph induced on 22 patients by the proposed model, MDMN. Each node represents one subject and the target domain is shown in blue. Note that although the target is only strongly connected to one source domain, the links between source domains allow them to share strength and make more robust predictions. The lines are labeled by the mean of directional weights learned in MDMN.

facilitate the mathematical and theoretical operations of pairwise matching in multiple domains. The underlying idea is also closely related to optimal transport for domain adaptation [7, 8], but address multiple domain matching.

The proposed method, MDMN, is visualized in Figure 1(b), compared with standard source to target matching scheme (Figure 1(a)), showing the matching of source domains. This tweak allows for already-similar domains to merge and share statistical strength, while keeping distant clusters of domains separate from one another. At test time, only the domains most relevant to the target are used [5, 32]. In essence, this induces a potentially sparse graph on all domains, which is visualized for 22 patients from one of our experiments in Figure 2. Any neural network architecture can be modified to use MDMN, which can be considered a stand-alone domain-matching module.

## 2    Method

Multiple Domain Matching Network (MDMN) is based upon the intuition that in the extracted feature space, inherently similar domains should have similar or identical distributions. By sharing strength within source domains, MDMN can better deal with the overfitting problem within each domain, a common problem in scientific domains. Meanwhile, the relationships between domains can also be learned, which is of interest in addition to classification performance.

In the following, suppose we are given $N$ observations $\{(\boldsymbol{x}_i, y_i, s_i)\}_{i=1}^N$ from $S$ domains, where $y_i$ is the desired label for $\boldsymbol{x}_i$ and $s_i$ is the domain. (In the target domain, the label $y$ is not provided and will instead be predicted.) **For brevity, we assume source domains are** $1, 2, \cdots, S-1$**, and the $S$th domain is the single target domain.** In fact, our approach works analogously for any number of unlabeled target domains.

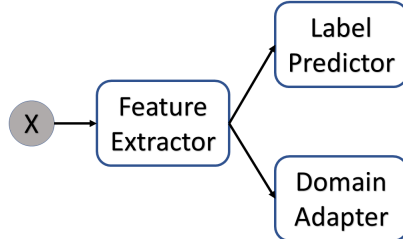

Figure 3: The framework of MDMN.

The whole framework, shown in Figure 3, is composed of an feature extractor (or encoder), a domain adapter (Sec. 2.1) and a label classifier (Sec. 2.2). In this work, we instantiate all three as neural networks. The encoder $E$ maps data points $\boldsymbol{x}$ to feature vectors $E(\boldsymbol{x})$. These features are then used by the label classifier to make predictions for the supervised task. They are also used by the domain adapter, encouraging extracted features $E(\boldsymbol{x})$ to be similar across nearby domains.

### 2.1    Domain Adaptation with Relationship Extraction

This section details the structure of the domain adapter. In order to adapt one domain to the others, one approach is to consider a penalty proportional to the distance between each distribution and the weighted mean of the rest. Specifically, let $P_s$ be the distribution over data points $\boldsymbol{x}$ in domain $\mathcal{D}_s$, and $P_{/s} = \frac{1}{S-1} \sum_{s'=1}^S w_{ss'} P_{s'}$ the distribution of data from all other domains $\mathcal{D}_{/s}^{w_s}$. Note that the weight $\boldsymbol{w}_s = [w_{s1}, \cdots, w_{sS}]$ is domain specific and $\boldsymbol{w}_s \in \mathbb{R}^S$, where $\boldsymbol{w}_s$ lies on the simplex with $||\boldsymbol{w}_s||_1 = 1$, $w_{ss'} \geq 0$ for $s' = 1, \ldots, S$ and $w_{ss} = 0$, which will be learned in the framework. In the following, we will use $\mathcal{D}_s$ to represent for its distribution $P_s$ in order to simplify the notation. Then we can encourage all domains to be close together in the feature space by adding the following term to the loss:

$$\mathcal{L}_D(E(\boldsymbol{x}; \boldsymbol{\theta}_E); \boldsymbol{\theta}_D) = \sum_{s=1}^S \beta_s d(\mathcal{D}_s, \mathcal{D}_{/s}^{w_s}), \tag{1}$$

where $d(\cdot, \cdot)$ is a distance between distributions (domains). Here it is used to measure the discrepancy between one domain and a weighted average of the rest. We assume the weight $\beta_s$ equals $\frac{1}{S-1}$ for $s = 1, \cdots, S-1$ and $\beta_S = 1$ to balance the penalty for the source and target domains, although this may be chosen as a tuning parameter. $\mathcal{L}_D$ is the total domain adapter loss function.

For the rest of this manuscript, we have chosen to use the Wasserstein distance as $d(\cdot, \cdot)$. This approach is facilitated by the use of Kantorovich-Rubenstein dual formulation of the Wasserstein-1 distance [2], which is given for distributions $\mathcal{D}_1$ and $\mathcal{D}_2$ as $d(\mathcal{D}_1, \mathcal{D}_2) = \sup_{||f||_L \leq 1} \mathbb{E}_{\boldsymbol{x} \sim P_1}[f(E(\boldsymbol{x}))] - \mathbb{E}_{\boldsymbol{x} \sim P_2}[f(E(\boldsymbol{x}))]$, where $||f||_L \leq 1$ denotes that the Lipschitz constant of the function $f(\cdot)$ is at

most 1, i.e. $|f(\boldsymbol{x}') - f(\boldsymbol{x})| \leq ||\boldsymbol{x}' - \boldsymbol{x}||_2$. $f()$ is any Lipschitz-smooth nonlinear function, which can be approximated by a neural network [2]. When $S$ is reasonably small ($< 100$), it is feasible to include $S$ small neural networks $f_s(\cdot; \boldsymbol{\theta}_D)$ to approximate these distances for each domain. In our implementation, we use shared layers in the domain adapter to enhance computational efficiency and the output of the domain adapter is $\boldsymbol{f}(\cdot; \boldsymbol{\theta}_D) = [f_1, \cdots, f_s, \cdots, f_S]$. The domain loss term is then given as

$$\sum_{s=1}^{S} \sup_{||f_s||_L \leq 1} \lambda_s \left( \mathbb{E}_{\boldsymbol{x} \sim \mathcal{D}_s}[f_s(E(\boldsymbol{x}))] - \mathbb{E}_{\boldsymbol{x} \sim \mathcal{D}_{/s}^{w_s}}[f_s(E(\boldsymbol{x}))] \right). \tag{2}$$

To make the domain penalty in (2) feasible, it is necessary to discuss how the penalty can be included in the optimization flow of neural network training. To develop this mathematical approach, let $\pi_s$ be the proportion of the data that comes from the $s^{th}$ domain, then the penalty can be rewritten as

$$
\begin{aligned}
& \frac{1}{S} \sum_{s=1}^{S} \beta_s \left( \mathbb{E}_{\boldsymbol{x} \sim \mathcal{D}_s}[f_s(E(\boldsymbol{x}))] - \mathbb{E}_{\boldsymbol{x} \sim \mathcal{D}_{/s}^{w_s}}[f_s(E(\boldsymbol{x}))] \right) \\
=\ & \mathbb{E}_{s \sim \text{Uniform}(1,\ldots,S)} \left[ \mathbb{E}_{\boldsymbol{x} \sim D_s}[\boldsymbol{r}_s^T \boldsymbol{f}(E(\boldsymbol{x}))] \right] \\
=\ & \mathbb{E}_{s \sim \boldsymbol{\pi}} \left[ \mathbb{E}_{\boldsymbol{x} \sim \mathcal{D}_s}[\frac{1}{S\pi_s} \times \boldsymbol{r}_s^T \boldsymbol{f}(E(\boldsymbol{x}))] \right],
\end{aligned}
\tag{3}
$$

where $\boldsymbol{f}(E(\boldsymbol{x}))$ is the concatenation of $f_s(E(\boldsymbol{x}))$, i.e. $\boldsymbol{f}(E(\boldsymbol{x})) = [f_1(E(\boldsymbol{x})), \cdots, f_S(E(\boldsymbol{x}))]^T$. $\boldsymbol{r} \in \mathbb{R}^S$ is defined as

$$\boldsymbol{r}_s = \begin{cases} -\beta_s w_{ss'}, & s' \neq s \\ \beta_s, & s' = s \end{cases}, s' = 1, \cdots, S. \tag{4}$$

The form in (3) is natural to include in an optimization loop because the expectation is empirically approximated by a mini-batch of data. Let $\{(\boldsymbol{x}_i, s_i)\}$, $i = 1, \ldots, N$ denote observations and their associated domain $s_i$, and then

$$\mathbb{E}_{s \sim \boldsymbol{\pi}} \left[ \mathbb{E}_{\boldsymbol{x} \sim \mathcal{D}_s}[\frac{1}{S\pi_s} \times \boldsymbol{r}_s^T \boldsymbol{f}(E(\boldsymbol{x}))] \right] \simeq \frac{1}{SN} \sum_{i=1}^{N} \pi_{s_i}^{-1} \boldsymbol{r}_{s_i}^T \boldsymbol{f}(E(\boldsymbol{x}_i; \boldsymbol{\theta}_E); \boldsymbol{\theta}_D). \tag{5}$$

The weight vector $\boldsymbol{w}_s$ for domain $\mathcal{D}_s$ should choose to focus only on relevant domains, and the weights on mismatched domains should be very small. As noted previously, adding uncorrelated domains hurts generalization performance [32]. In our Theorem 3.3, we shows that a weighting scheme with these properties decreases the target error bound. Once the function $f_s(\cdot; \boldsymbol{\theta}_D)$ is known, we can estimate $\boldsymbol{w}_s$ by using a softmax transformation on the function expectations from $f_s$ between any two domains. Specifically, the weight $\boldsymbol{w}_s$ to match $\mathcal{D}_s$ to other domains is calculated as

$$\boldsymbol{w}_s = \text{softmax}_{/s}(\kappa \boldsymbol{l}_s), \text{ with } l_{ss'} = \left( \mathbb{E}_{\boldsymbol{x} \sim \mathcal{D}_s}[f_s(E(\boldsymbol{x}))] - \mathbb{E}_{\boldsymbol{x} \sim \mathcal{D}_{s'}}[f_s(E(\boldsymbol{x}))] \right), \tag{6}$$

where $\boldsymbol{l}_s = [l_{s1}, \cdots, l_{ss'}, \cdots, l_{sS}]$. The subscript $/s$ means that the value $w_{ss}$ is restricted to 0 and $l_{ss}$ is excluded from the softmax. The scalar quantity $\kappa$ controls how peaked $\boldsymbol{w}_s$ is. Note that setting $\boldsymbol{w}_s$ in (2) to the closest domain and 0 otherwise would correspond to the $\kappa \to \infty$ case, and $\kappa \to 0$ corresponds to an unweighted (e.g. conventional) case. It is beneficial to force the domain regularizer to match to multiple, but not necessarily all, available domains. Practically, we can either modify $\kappa$ in the softmax or change the Lipschitz constant used to calculate the distance (as was done). As an example, the learned graph connectivity is shown in Figure 2 is constructed by thresholding $\frac{1}{2}(w_{ss'} + w_{s's})$ to determine connectivity between nodes.

## 2.2 Combining the Loss Terms

The proposed method uses the loss in (5) to perform the domain matching. A label classifier is also necessary, which is defined as a neural network parameterized by $\boldsymbol{\theta}_Y$. The label classifier in Figure 3 is represented as $Y[E(\boldsymbol{x})]$, where the classifier $Y$ is applied on the extracted feature vector $E(\boldsymbol{x})$. The label predictor usually contains several fully connected layers with nonlinear activation functions. The cross entropy loss is used for classification, i.e. $\mathcal{L}_Y(\boldsymbol{x}, \boldsymbol{y}; \boldsymbol{\theta}_Y, \boldsymbol{\theta}_E) = \sum_{i=1}^{N} \sum_{c=1}^{C} y_{ic} \log Y_c[E(\boldsymbol{x}_i)]$, where $Y_c$ means the $c^{th}$ entry of the output. The MSE loss is used for regression.

With the label prediction loss $\mathcal{L}_Y$, the complete network loss is given by

$$\min_{\boldsymbol{\theta}_E, \boldsymbol{\theta}_Y} \max_{\boldsymbol{\theta}_D} \mathcal{L}_Y(\boldsymbol{\theta}_Y, \boldsymbol{\theta}_E) + \rho \mathcal{L}_D(\boldsymbol{\theta}_D, \boldsymbol{\theta}_E), \tag{7}$$

where $\boldsymbol{\theta}_E$ denotes the parameters in the feature extractor/encoder, $\boldsymbol{\theta}_D$ denotes the parameters in the domain adapter, and $\boldsymbol{\theta}_Y$ in the label classifier. The pseudo code for training is given in Algorithm 1.

---

**Algorithm 1** Multiple Source Domain Adaptation via WDA

---

**Input**: Source samples from $\mathcal{D}_s$, $s = 1, \cdots, S - 1$ and target samples from $\mathcal{D}_S$. Note that we assume index $1, \cdots, S - 1$ are for source domains and $S$ is for the target domain. Iteration $k_Y$ and $k_D$ for training label classifier and domain discriminator.
**Output**: Classifier parameters $\boldsymbol{\theta}_E, \boldsymbol{\theta}_Y, \boldsymbol{\theta}_D$.

---

**for** $iter = 1$ to $max\_iter$ **do**
    Sample a mini-batch of $\{\boldsymbol{x}^s\}$ from $\{\mathcal{D}_s\}_{s=1}^{S-1}$ and $\{\boldsymbol{x}^t\}$ from $\mathcal{D}_S$.
    **for** $iter_Y = 1$ to $k_Y$ **do**
        Compute $l_{ss'} = \mathbb{E}_{\boldsymbol{x} \in \mathcal{D}_s}\left[f_s(E(\boldsymbol{x}))\right] - \mathbb{E}_{\boldsymbol{x} \in \mathcal{D}_{s'}}\left[f_s(E(\boldsymbol{x}))\right]$ for $\forall s, s' \in [1, S]$.
        Compute the weight vectors $\boldsymbol{w}_s = \mathrm{softmax}_{/s}(\boldsymbol{l}_s)$ and $w_{ss} = 0$ for $\forall s \in [1, S]$, where
        $\boldsymbol{l}_s = (l_{s1}, \cdots, l_{sS})$.
        Compute domain loss $\mathcal{L}_D^W(\boldsymbol{x}^s, \boldsymbol{x}^t)$ and classifier loss $\mathcal{L}_Y(\boldsymbol{x}^s)$.
        Compute $\nabla \boldsymbol{\theta}_Y = \frac{\partial \mathcal{L}_Y}{\partial \boldsymbol{\theta}_Y}$ and $\nabla \boldsymbol{\theta}_E = \frac{\partial \mathcal{L}_Y}{\partial \boldsymbol{\theta}_E} + \rho \frac{\partial \mathcal{L}_D}{\partial \boldsymbol{\theta}_E}$
        Update $\boldsymbol{\theta}_Y = \boldsymbol{\theta}_Y - \nabla \boldsymbol{\theta}_Y$, $\boldsymbol{\theta}_E = \boldsymbol{\theta}_E - \nabla \boldsymbol{\theta}_E$.
    **end for**
    **for** $iter_D = 1$ to $k_D$ **do**
        Update the weight vectors $\boldsymbol{w}_s, \forall s \in [1, S]$.
        Compute $\mathcal{L}_D(\boldsymbol{x}^s, \boldsymbol{x}^t)$ and $\nabla \boldsymbol{\theta}_D = \frac{\partial \mathcal{L}_D}{\partial \boldsymbol{\theta}_D}$.
        Update $\boldsymbol{\theta}_D = \boldsymbol{\theta}_D + \nabla \boldsymbol{\theta}_D$.
    **end for**
**end for**

---

During training, the target domain weight $\beta_S$ in eq. (1) is always set to one, while sources domain weights are normalized to have sum one. This is because the ultimate goal is to work well on target domain. We use the gradient penalty introduced in [18] to implement the Lipschitz constraint. A concern is that the feature scale may change and impact the Wasserstein distance. One potential solution to this is to include batch normalization to keep the summary statistics of the extracted features constant. In practice, this is not necessary. Adam [20] is used as the optimization method while the gradient descent step in Algorithm 1 reflects the basic strategy.

### 2.3 Complexity Analysis

Although the proposed algorithm computes pairwise domain distance, the computational cost in practice is similar compared to standard DANN model.

For the domain loss functions, we share all the bottom layers for all domains. This is similar to the setup of a multi-class domain classifier with softmax output while in our model, the output is a real number. Specifically, the pairwise distance (6) is updated in each mini-batch by averaging samples in the same domain.

$$\hat{l}_{ss'} \approx \frac{1}{n_s} \sum_{\forall \boldsymbol{x}_i \in \mathcal{D}_s} f_s(E(\boldsymbol{x}_i)) - \frac{1}{n_{s'}} \sum_{\forall \boldsymbol{x}_i \in \mathcal{D}_{s'}} f_s(E(\boldsymbol{x}_i)) \tag{8}$$

Because these pairwise calculations happen late in the network, their computational cost is dwarfed by feature generation. We believe that the methods will easily scale to hundreds of domains based on computational and memory scaling. We use exponential smoothing during the updates to improve the quality of the estimates, with $l_{ss'}^{t+1} = 0.9 l_{ss'}^t + 0.1 \hat{l}_{ss'}^c$. $\hat{l}_{ss'}^c$ is the value from current iteration's mini-batch. Then the softmax is applied on the calculated values to get the weight $w_{ss'}$. This procedure is used to update $\boldsymbol{w}_s$, so those parameters are not included in the backpropagation. The domain weights and network parameters are updated iteratively, as shown in Algorithm 1.

## 3 Theoretical Results

In this section, we investigate the theorems and derivations used to bound the target error with the given method in Section 2. Specifically, the target error is bounded by the source error, the source-target distance plus additional terms which is constant under certain data and hypothesis

classes. The theory is developed based on prior theories of source to target adaptation. The adaptation within source domains can be developed in the same way. Additional details and derivations are available in the Supplemental Section A.

Let $\mathcal{D}_s$ for $s = 1, \cdots, S$ and $\mathcal{D}_T$ represent the source and the target domain, respectively. Note that there is a notation change in the target domain, where the $S$th domain was denoted as the target in previous section. Here, it is easier to separate the target domain out. Suppose that there is probabilistic true labeling functions $g_s$, $g_T : \mathcal{X} \to [0, 1]$ and a probabilistic hypothesis $f : \mathcal{X} \to [0, 1]$, which in our case is a neural network. The output value of the labeling function determines the probability that the sample is 0 or 1. $g_s$, $g_T$ are assumed Lipschitz smooth with parameters $\lambda_s$ and $\lambda_T$, respectively. This differs from the previous derivation [14] that assumes that the hypothesis and labeling function were deterministic ($\{0, 1\}$). In the following, the notation of encoder E() is removed for simplicity. Thus $f(\boldsymbol{x})$ is actually $f(E(\boldsymbol{x}; \boldsymbol{\theta}_E); \boldsymbol{\theta}_D)$. Since we first only focus on the adaptation from source to target, the output of $f(\cdot)$ in this section is a scalar (The last element of $\boldsymbol{f}(\cdot)$). Same for notation $w_s$, which is the domain similarity of $\mathcal{D}_s$ and target.

**Definition 3.1** (Probabilistic Classifier Discrepancy). *The probabilistic classifier discrepancy for domain $\mathcal{D}_s$ is defined as*

$$\gamma_s(f, g) = \mathbb{E}_{\mathbf{x} \sim \mathcal{D}_s}[|f(\mathbf{x}) - g(\mathbf{x})|]. \tag{9}$$

Note that if the label hypothesis is limited to $\{0, 1\}$, this is classification accuracy. In order to construct our main theorem, we use notation $||f||_L \leqslant \lambda$ to denote $\lambda$-smooth function. Mathematical details are given in Definition A.6 in the appendix. Next we define the weighted Wasserstein-like quantity between sources and the target.

**Definition 3.2** (Weighted Wasserstein-like quantity). *Given $S$ multi-source probability distributions $P_s$, $s = 1, \cdots, S$ and $P_T$ for the target domain, the difference between the weighted source domains $\{\mathcal{D}_s\}_{s=1}^S$ and target domain $\mathcal{D}_T$ is described as,*

$$\alpha_\lambda(\mathcal{D}_T, \textstyle\sum_s w_s \mathcal{D}_s) = \max_{f:\mathcal{X} \to [0,1], ||f||_L \leq \lambda} \mathbb{E}_{\boldsymbol{x} \sim \mathcal{D}_T}[f(\mathbf{x})] - \mathbb{E}_{\boldsymbol{x} \sim \sum_s w_s \mathcal{D}_s}[f(\mathbf{x})]. \tag{10}$$

Note that if the bound on the function from 0 to 1 is removed, then this quantity is the Kantorovich-Rubinstein dual form of the Wasserstein-1 distance. As $\lambda \to \infty$, this is the same as the commonly used $L1$-divergence or variation divergence [4]. Thus, we can derive this theorem with $\mathcal{H}$-divergence exactly, but prefer to use the smoothness constraint to match the used Wasserstein distance. We also define $f^*$ as an optimal hypothesis that achieves the minimum discrepancy $\gamma^*$, which is given in the appendix A.3. Now we come to the main theorem of this work.

**Theorem 3.3** (Bound on weighted multi-source discrepancy). *For a hypothesis $f : \mathcal{X} \to [0, 1]$,*

$$\gamma_T(f, g_T) \leq \textstyle\sum_{s=1}^S w_s \gamma_s(f, g_s) + \alpha_{\lambda_T + \lambda^*}(\sum_{s=1}^S w_s \mathcal{D}_s, \mathcal{D}_T) + \gamma^* \tag{11}$$

The quantity $\gamma^*$ given in (27) is defined in the appendix and addresses the fundamental mismatch in true labeling functions, which is uncontrollable by domain adaptation. Note that a weighted sum of Lipschitz continuous functions is also Lipschitz continuous. $\lambda^*$ is the Lipschitz continuity for the weighted domain combination $\lambda^* = \sum_{s=1}^S w_s \lambda_s$, where $f_s()$ of domain $\mathcal{D}_s$ has Lipschitz constant $\lambda_s$. We note that in Theorem 3.3 we are dependent on the weighted sum of the source domains, implying that increasing the weight on irrelevant source domain may *hurt* generalization performance. This matches existing literature. Second, a complex model with high learning capacity will reduce the source error $\gamma_s(f, g_s)$, but the uncertainty introduced by the model will increase the domain discrepancy measurement $\alpha_{\lambda + \lambda^*}(\{\mathcal{D}_s\}_{s=1}^S, \mathcal{D}_T)$.

Compared to the multi-source domain adversarial network's (MDAN's) [45] bound, $\gamma_T(f, g_T) \leq \max_s \gamma_s(f, g_s) + \max_s d_{\mathcal{H}\Delta\mathcal{H}}(\mathcal{D}_s, \mathcal{D}_T) + \gamma^*$, where the definition of $d_{\mathcal{H}\Delta\mathcal{H}}$ is given in appendix section A.2. Theorem 3.3 reveals that weighting has a tighter bound because an irrelevant domain with little weight will not seriously hurt the generalization bound whereas prior approaches have taken the max over the least relevant domain. Also, the inner domain matching helps prevent spurious relationships between irrelevant domains and the target. Therefore, MDAN can pick out more relevant source domains compared to the alternative methods evaluated.

# 4 Related Work

There is a large history in domain adaptation to transfer source distribution information to the target distribution or vice versa, and has been approached in a variety of manners. Kernel Mean

Matching (KMM) is widely used in the assumption that target data can be represented by a weighted combination of samples in the source domain [37, 19, 12, 29, 40]. Clustering [25] and late fusion [1] approaches have also been evaluated. Distribution matching has been explored with the Minimum Mean Discrepancy [29] and optimal transport [8, 7], which is similar to the motivation used in our domain penalization.

With the increasing use of neural networks, weight sharing and transfer has emerged as an effective strategy for domain adaptation [15]. With the development of Generative Adversarial Networks (GANs) [17], adversarial domain adaptation has become popular. The Domain Adversarial Neural Network (DANN) is a newly proposed model for feature adaptation rather than simple network weight sharing [14]. Since its publication, the DANN approach has been generalized [39, 43] and extended to multiple domains [45]. In the multiple domain case, a weighted combination of source domains is used for adaptation. [22] is based on the DANN framework, but uses distributional summary statistics in the adversary. Several other methods use source or target sample generation with GANs on single source domain adaptation [35, 27, 26, 33], but extensions to multi-source domains are not straightforward. [3] provides a multi-stage multi-source domain adaptation.

There has also been theoretical analysis of error bounds for multi-source domain adaptation. [9] analyzes the theory on distributed weighted combining of multiple source domains. [32] gives a bound on target loss when only using $k$-nearest source domains. It shows that adding more uncorrelated source domains training data hurts the generalization bound. The bound that [4] gives is also on the target risk loss. It introduces the $\mathcal{H}$-divergence as a measurement of the distance between source and target domains. [5] further analyzes whether source sample quantity can compensate for quality under different methods and different target error measurements.

Domain adaptation can be used in a wide variety of applications. [16, 10] uses it for natural language processing tasks. [12] perform video concept detection using multi-source domain adaptation with auxiliary classifiers. [15, 14, 1, 3, 39] focus on image domain transfer learning. The multi-source domain adaptation in previous works is usually limited to fewer than five source domains. Some scientific applications have more challenging situation by adapting from a significantly higher number of source domains [44]. In some neural signals, different methods have been employed to transfer among subjects based on hand crafted EEG features [38, 24]; however, these models need to be trained in several steps, making them less robust.

## 5 Experiment

We tested MDMN by applying it to three classification problems: image recognition, natural-language sentiment detection, and multi-channel time series analysis. The sentiment classification task is given in the Appendix due to limited space.

### 5.1 Results on Image Datasets

We first test the performance of the proposed MDMN model on MNIST, MNISTM, SVHN and USPS datasets. Visualizations of these datasets are given in the Appendix Section C.1. In each test, one dataset is left out as target domain while the remaining three are treated as source domains. The feature extractor $E$ consists of two convolutional layers plus two fully connected layers. Both the label predictor and domain adapter are two layers MLP. ReLU nonlinearity is used between layers. The baseline method is the concatenation of feature extractor and label predictor as a standard CNN but it has no access to any target domain data during training process.

While TCA [34] and SA [13] methods can process raw images, the results are significantly stronger following a feature extraction step. The results from these methods are given by following two independent steps. First, a convolutional neural network with the same structure as in our proposed approach is used as a baseline. This model is trained on the source domains, and then features are extracted for all domains to use as inputs into TCA and SA. Another issue is computational complexity for TCA, because this algorithm computes the matrix inverse during the inference, which is of complexity $\mathcal{O}(N^3)$. Hence, data was limited for this algorithm. For the adversarial based algorithms [39, 14, 45] and MDMN model, the domain classifier is the uniform, which is a two layer MLP with ReLU nonlinearities and a soft-max top layer.

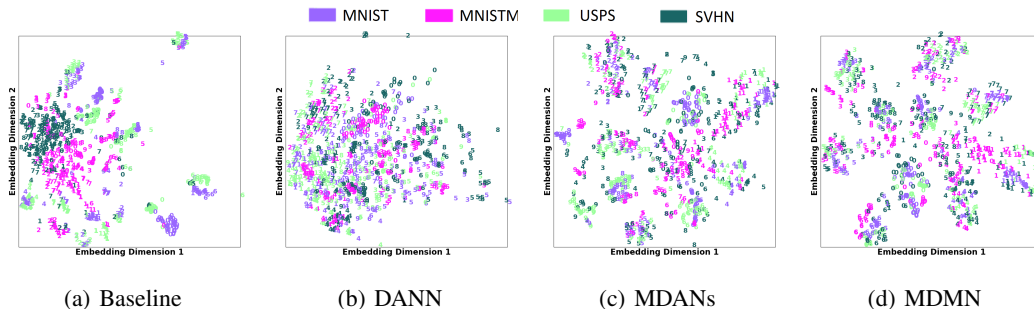

(a) Baseline       (b) DANN       (c) MDANs       (d) MDMN

Figure 4: Visualization of feature spaces of different models by t-SNE. Each color represents one dataset of MNIST, MNISTM, SVHN and USPS. The testing target domain is MNISTM. The digit label is shown in the plot. The goal is to adapt generalized feature from source domains to the target domains; the digits should cluster together rather than the color clustering.

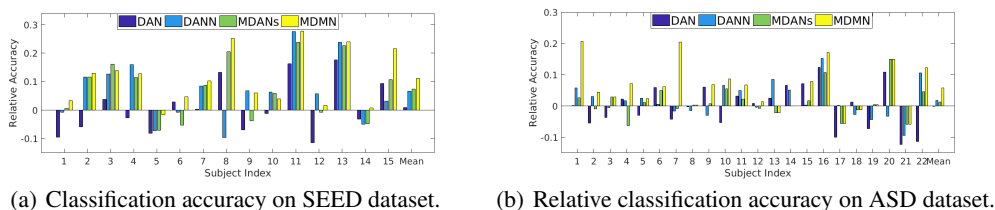

(a) Classification accuracy on SEED dataset.       (b) Relative classification accuracy on ASD dataset.

Figure 5: Relative classification accuracy by subject on two EEG datasets. The accuracy without subtracting the baseline performance is given in appendix C.2.

The classification accuracy is compared in Table 1. The top row shows the baseline result on the target domain with the classifier trained on the three other datasets. The proposed model MDMN outperforms the other baselines on all datasets. Note that some domain-adaptation algorithms actually lower the accuracy, revealing that domain-specific features are being transfered. Another problem encountered is the mismatch between the source domain and target domain. For instance, when the target domain is the MNIST-M dataset, it is expected to give large weight to MNIST dataset samples during training. However, algorithms like TCA, SA and DANN equally weight all source domain datasets, making the result worse than MDMN.

If we project the feature vector for each data to two dimensions using the TSNE embedding [31], the features are shown in Figure 4. The goal is to mix different colors while distinguishing different digits. The baseline model in Figure 4(a) shows no adaptation for the target domain, i.e. the digit '0' from USPS and MNIST datasets form two islands if domain adaptation is not imposed. The DANN model and the MDANs model shows some "mixing" effect, which indicates that

| Acc. % | MNIST | MNISTM | USPS | SVHN |
|---|---|---|---|---|
| Baseline | 94.6 | 60.8 | 89.4 | 43.7 |
| TCA [34] | 78.4 | 45.2 | 75.4 | 39.7 |
| SA [13] | 90.8 | 59.9 | 86.3 | 40.2 |
| DAN [28] | 97.1 | 67.0 | 90.4 | 51.9 |
| ADDA [39] | 89.0 | 80.3 | 85.2 | 43.5 |
| DANN [14] | 97.9 | 68.8 | 89.3 | 50.1 |
| MDANs [45] | 97.2 | 68.5 | 90.1 | 50.5 |
| MDMN | **98.0** | **83.8** | **94.5** | **53.1** |

Table 1: Accuracy on image classification. For the TCA method, 20% of the data was randomly selected.

domain adaptation is happening because the extracted features are more similar between domains. MDMN has the most clear digit mixing effect. The model finds the digit label features instead of domain specific features. A larger figure of the same result is given in Appendix C.1 for enhanced clarity.

## 5.2    Result on EEG Time Series

Two datasets are used to evaluate performance on Electroencephalography (EEG) data: SEED dataset and an Autism Spectrum Disorder (ASD) dataset.

The **SEED** dataset [46] focuses on analyzing emotion using EEG signal. This dataset has 15 subjects. The EEG signal is recorded when each subject watches 15 movie clips for 3 times at three different days. Each video clip is attached with a negative/neutral/positive emotion label. The sampling rate is at $1000Hz$ and a 62-electrode layout is used. In our experiment, we downsample the EEG signal to $200Hz$. The test scheme is the leave-one-out cross-validation. In each time, one subject is picked out as test and the remaining 14 subjects are used as training and validation.

The **Autism Spectrum Disorder** (ASD) dataset [11] aims at discovering whether there are significant changes in neural activity in a open label clinical trial evaluating the efficacy of a single infusion of autologous cord blood for treatment of ASD [11]. The study involves 22 children from ages 3 to 7 years undergoing treatment for ASD with EEG measurements at baseline (T1), 6 months post treatment (T2), and 12 months post treatment (T3). The signal was recorded when a child was watching a total of three one-minute long videos designed to measure responses to dynamic social and nonsocial stimuli. The data has 121 signal electrodes. The classification task is to predict the treatment stage T1, T2 and T3 to test the effectiveness of the treatment and analyze what features are dynamic in response to the treatment. By examining the features, we can track how neural changes correlate to this treatment stages. We also adopt the leave-one-out cross-validation scheme for this dataset, where one subject is left out as testing, the remaining 21 subjects are separated as training and validation. Leaving complete subjects out better estimates generalization to a population in these types of neural tasks [42].

The classification accuracy using different methods is compared in Table 2. In this setting, we choose our baseline model as the SyncNet [23]. SyncNet is a neural network with structured filter targeting at extracting neuroscience related features. The simplest framework of SyncNet is adopted which only contains one layer of convolutional filters. As in [23], we set the filter number to 10 for both datasets. For TCA, SA and ITL methods, the baseline model was trained as before without a domain adapter on the source domain data. Extracted features from this model were then used to extract features from target domains.

MDMN outperforms other competitors on both EEG datasets. A subject by subject plot is shown in Figure 5. Because performance on subjects is highly variable, we only visualize performance relative to baseline, and absolute performance is visualized in Figure 8 in the appendix. Because the source domains are large but each source domain is highly variable, the requirement to find relevant domains is of increased importance on both of the EEG datasets. For the ASD dataset, DANN and MDANs do not match the performance of MDMN mainly because they cannot correctly pick out most related subject from source domains. This is also true for TCA,

| Dataset | SEED | ASD |
|---|---|---|
| SyncNet [23] | 49.29 | 62.06 |
| TCA [34] | 39.70 | 55.65 |
| SA [13] | 53.90 | 62.53 |
| ITL [36] | 45.27 | 54.62 |
| DAN [28] | 50.28 | 61.88 |
| DANN [14] | 55.87 | 63.81 |
| MDANs [45] | 56.65 | 63.38 |
| MDMN | **60.59** | **67.78** |

Table 2: Classification mean accuracy in percentage on EEG datasets.

SA and ITL. Our proposed algorithm MDMN overcomes this problem by computing domain similarity in feature space while performing feature mapping, and a domain relationship graph by subject is given in Figure 2. Each subject is related to all the others with different weight. The missing edges, like the edges to node 's10', are those with weight less than $0.09$. Our algorithm automatically finds the relationship and the domain adaptation happens with the calculated weight, instead of treating all domains equally.

# 6   Conclusion

In this work, we propose the Multiple Domain Matching Network (MDMN) that uses feature matching across different source domains. MDMN is able to use pairwise domain feature similarity to give a weight to each training domain, which is of key importance when the number of source domains increases, especially in many neuroscience and biological applications. While performing domain adaptation, MDMN can also extract the relationship between domains. The relationship graph itself is of interest in many applications. Our proposed adversarial training framework further applies this idea on different domain adaptation tasks and shows state-of-the-art performance.

## Acknowledgements

Funding was provided by the Stylli Translational Neuroscience Award, Marcus Foundation, NICHD P50-HD093074, and NIMH 3R01MH099192-05S2.

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
