[Supplementary Material]

# A  Theoretical Analysis

## A.1  Preliminaries and Key Prior Analysis

In this section, we review key concepts from previous works, which will be used in our theorems and algorithm in Sections 2 and 3. The theorems mainly focus on bounding the target domain classification error by the source domain training error and an source-target domain discrepancy. The notations are the same as main text.

First, we will discuss the established theorems on binary labeling and hypothesis class, which is different from the continuous output introduced in the main article, and then discuss the changes introduced in our theorem. Let $\mathcal{D}_S$ and $\mathcal{D}_T$ represent the source and the target domain, respectively. Suppose that there are true labeling functions $g_S,\ g_T : \mathcal{X} \to \{0,1\}$ for source and target domain respectively, and a hypothesis $h : \mathcal{X} \to \{0,1\}$ which takes a point in the data space and give either 0 or 1 label. Here we briefly introduce some definitions from [4].

**Definition A.1** (Classification error). *Define the classification error as*

$$\epsilon_S(f,g) = \mathbb{E}_{\boldsymbol{x}\sim\mathcal{D}_S}[|f(\boldsymbol{x}) - g(\boldsymbol{x})|]. \tag{12}$$

**Definition A.2** (Variation Divergence). *A measure of divergence for distributions is the $L^1$ or variation divergence*

$$d_1(\mathcal{D},\mathcal{D}') = 2\sup_{B\in\mathcal{B}}|Pr_{\mathcal{D}}[B] - Pr_{\mathcal{D}'}[B']|, \tag{13}$$

*where $\mathcal{B}$ is the set of measurable subsets under two different domains $\mathcal{D}$ and $\mathcal{D}'$.*

The main theorem developed by [4] is the following, where the target domain error is bounded by three terms.

**Theorem A.3** (From Theorem 1 in [4]). *For a hypothesis $h$, the target error is bounded by*

$$\epsilon_T(f,g_T) \leq \epsilon_S(f,g_S) + d_1(\mathcal{D}_S,\mathcal{D}_T)+$$
$$\min\left\{\mathbb{E}_{\mathcal{D}_S}[|g_S(\mathbf{x}) - g_T(\mathbf{x})|], \mathbb{E}_{\mathcal{D}_T}[|g_S(\mathbf{x}) - g_T(\mathbf{x})|]\right\} \tag{14}$$

The proof of this was given previously by [4]. Note that the third term in the R.H.S. of (14) is uncontrollable by the chosen classification algorithm because it is fundamental disagreement between the true training and testing classifiers, but the first two terms of the R.H.S. are controllable by the chosen procedure. Next, we will give the definition of $\mathcal{H}$-divergence. Via the $\mathcal{H}$-divergence, this quantity can be estimated with a probabilistic bound via a trained classifier.

**Definition A.4** ($\mathcal{H}$-Divergence). *Given two domains $\mathcal{D}$ and $\mathcal{D}'$ with distribution $P$ and $P'$ over input $\boldsymbol{x}$. For a hypothesis class $\mathcal{H}$, the $\mathcal{H}$ divergence is defined as*

$$d_{\mathcal{H}}(\mathcal{D},\mathcal{D}') = 2\sup_{f\in\mathcal{H}}|Pr_{\boldsymbol{x}\sim P}[f(\boldsymbol{x}) = 1] - Pr_{\boldsymbol{x}\sim P'}[f(\boldsymbol{x}) = 1]|. \tag{15}$$

With the $\mathcal{H}$-Divergence, authors in [4] develop the bound using $d_{\mathcal{H}\Delta\mathcal{H}}$ distance. When using finite VC classes, $d_{\mathcal{H}\Delta\mathcal{H}}$ is superior to $d_1$ because it can be bounded by a finite sample approximation. However, their bound (Theorem 2 in [4]) requires the VC dimension of the hypothesis space $\mathcal{H}$. In our work, $\mathcal{H}$ is instantiated as a neural network. The extremely large VC dimension of a complex neural network complicates its application here.

## A.2  An Upper Bound for Two Domain Adaptation

In this section, we will give the two domain case where $\gamma^*$ in our main theorem 3.3 can be written out explicitly. In order to make this section self-complete, we repeat definition 3.1 again.

**Definition A.5** (Probabilistic Classifier Discrepancy). *The probabilistic classifier discrepancy is defined as*

$$\gamma_S(f,g) = \mathbb{E}_{\mathbf{x}\sim\mathcal{D}_S}[|f(\mathbf{x}) - g(\mathbf{x})|].$$

The difference between $\epsilon_S(f,g)$ and $\gamma_S(f,g)$ is in the function output space. Where in $\epsilon_S(f,g)$, the output of $f$ and $g$ are limited to binary, while in $\gamma_S(f,g)$, the output can be any real number in $[0,1]$. Instead of assuming that the labeling function and hypothesis are limited to a deterministic binary

response, suppose that a probability of the positive label is returned, such that $f, g_S, g_T : \mathcal{X} \to [0, 1]$. Note that the binary labeling and hypothesis are a subset of this set of continuous functions. Here we assume a probabilistic output, but an alternative approach would be to describe the hypothesis class as a deterministic function with "probabilistic" Lipschitz, as in [5].

**Definition A.6** (Lipschitz continuity). *A function $f : \mathcal{X} \to \mathbb{R}$ is said to be Lipschitz continuous with parameter $\lambda$ if for any vectors $\mathbf{x}, \mathbf{y} \in \mathcal{X}$ if*

$$|f(\mathbf{y}) - f(\mathbf{x})| \le \lambda ||\mathbf{y} - \mathbf{x}||_2. \tag{16}$$

In order to deal with probabilistic output, the Lipschitz constraint in Definition A.6 is to encourage the probabilistic estimates to be as correct as possible while encouraging smooth boundaries to facilitate optimal decision making. In many applications (e.g. signal processing), we know that the probabilities of the labeling must be *smooth* because the decision boundaries are smoothed by additive noise. Suppose that the hypothesis class of $f$ is limited to functions that are $\lambda_H$-smooth and that the true labeling functions are $\lambda$-smooth, then we can prove an alternative bound on the discrepancy.

**Definition A.7** (Wasserstein-like quantity). *Given two probability distributions $P$ and $P'$ for domain $\mathcal{D}$ and $\mathcal{D}'$, the difference between the two domains is described as,*

$$\alpha_\lambda(\mathcal{D}, \mathcal{D}') = \max_{f:\mathcal{X} \to [0,1], ||f||_L \le \lambda} \mathbb{E}_{\mathcal{D}}[f(\mathbf{x})] - \mathbb{E}_{\mathcal{D}'}[f(\mathbf{x})]. \tag{17}$$

*Note that if the bound on the function from 0 to 1 is removed, then this quantity is the Kantorovich-Rubinstein dual form of the Wasserstein-1 distance.*

Next, we will give more explanation of the Wasserstein-like quantity from Definition A.7. Remember that $||f||_L$ defines the Lipschitz continuity of the function. As a reminder, the Kantorovich-Rubinstein dual form of the Wasserstein-1 distance [41] is

$$W_1(\mathcal{D}, \mathcal{D}') = \max_{f:\mathcal{X} \to \mathbb{R}, ||f||_L \le 1} \mathbb{E}_{\mathcal{D}}[f(\mathbf{x})] - \mathbb{E}_{\mathcal{D}'}[f(\mathbf{x})]. \tag{18}$$

$W_1(\mathcal{D}, \mathcal{D}')$ with respect to a smoothness of $\lambda$ instead of 1 would just result in $\lambda W_1(\mathcal{D}, \mathcal{D}')$. Therefore, the only difference between $\alpha$ and $W_1$ is the bound on the output values of the function, and

$$\frac{1}{\lambda} \alpha_\lambda(\mathcal{D}, \mathcal{D}') \le W_1(\mathcal{D}, \mathcal{D}').$$

Further, note that

$$\alpha_\lambda(\mathcal{D}, \mathcal{D}') \le d_1(\mathcal{D}, \mathcal{D}'), \tag{19}$$

which states that this $\alpha$-quantity is tighter than the corresponding variation divergence used by [4]. $d_1()$ is defined in definition A.2. This is expected because we have assumed an additional constraint on the system.

Now we are ready for the main theorem of this work. In order to derive the theorem, we assume that the true labeling functions $g_S$ and $g_T$ are $\lambda$-smooth and that the proposed hypothesis class $f$ is $\lambda_H$-smooth.

**Theorem A.8** (Bound on discrepancy). *For a hypothesis $f : \mathcal{X} \to [0, 1]$,*

$$\gamma_T(f, g_T) \le \gamma_S(f, g_S) + \alpha_{\lambda+\lambda_H}(\mathcal{D}_S, \mathcal{D}_T) + \tag{20}$$
$$\min \left\{ \mathbb{E}_{\mathcal{D}_S}[|g_S(\mathbf{x}) - g_T(\mathbf{x})|], \mathbb{E}_{\mathcal{D}_T}[|g_S(\mathbf{x}) - g_T(\mathbf{x})|] \right\}$$

This theorem gives an upper bound of the target discrepancy $\gamma_T(f, g_T)$ in terms of three quantities. The first is the classifier discrepancy on the source domain, $\gamma_S(f, g_s)$. The third term refers to fundamentally how mismatched the true labels for the two domains are, which is not addressable by domain adaptation. The fundamental difference between our theorem and the theorem of [4] is that our second term incorporates the Wasserstein-like quantity, whereas the equivalent term in the existing theorem is approximated by an $\mathcal{H}$-divergence. When our assumptions are satisfied, our theorem is *tighter* than the existing theorem, motivating the use of the Wasserstein distance in our domain penalty. The Wasserstein was chosen in lieu of the form in (17) because of the existing approximation techniques utilized in our computational methods.

*Proof.* We start with a statement of equivalence:

$$\gamma_T(f, g_T) = \gamma_T(f, g_T) + \gamma_S(f, g_S) - \gamma_S(f, g_S) + \gamma_S(f, g_T) - \gamma_S(f, g_T) \qquad (21)$$

and then bound the output term by taking the absolute value of differences:

$$
\begin{aligned}
\gamma_T(f, g_T) &\leq \gamma_S(f, g_S) + |\gamma_S(f, g_T) - \gamma_S(f, g_S)| + |\gamma_T(f, g_T) - \gamma_S(f, g_T)| \\
&= \gamma_S(f, g_S) + \mathbb{E}_{\mathcal{D}_S}[|g_S(\mathbf{x}) - g_T(\mathbf{x})|] + |\gamma_T(f, g_T) - \gamma_S(f, g_T)|.
\end{aligned}
$$

We note that an alternative approach to this derivation would be to use a deterministic hypothesis space and a probabilistic Lipschitzness assumption [5], but this is left to future work.

The first two terms proceed exactly as by [4]; further derivations are not provided. However, the third term of the right hand side differs. Focusing on the last term, let $P_S$ and $P_T$ be the densities of $\mathcal{D}_S$ and $\mathcal{D}_T$, respectively. Then,

$$|\gamma_T(f, g_T) - \gamma_S(f, g_T)| \leq \left| \int (P_T(\mathbf{x}) - P_S(\boldsymbol{x})) |f(\boldsymbol{x}) - g_T(\boldsymbol{x})| d\boldsymbol{x} \right|$$

Note that because of the additivity properties of Lipschitz smooth functions, $h(\boldsymbol{x}) = f(\mathbf{x}) - g_T(\mathbf{x})$ is at least $\lambda + \lambda_H$ smooth. Second, note that the absolute value does not change the Lipschitz constant. Third, note that the $f(\boldsymbol{x})$ is bounded on $[0, 1]$ and $-f_T(\boldsymbol{x})$ is bounded on $[-1, 0]$, but that the absolute value of a function with range $[-1, 1]$ has range $[0, 1]$. Therefore, the previous equation can be bounded by,

$$|\gamma_T(h, f_T) - \gamma_S(h, f_T)| \leq \left| \max_{h : \mathcal{X} \to [0,1], ||h|| \leq \lambda + \lambda_H} \int (P_T(\mathbf{x}) - P_S(\mathbf{x})) h(\mathbf{x}) d\mathbf{x} \right|, \qquad (22)$$

$$= \left| \max_{h : \mathcal{X} \to [0,1], ||h|| \leq \lambda + \lambda_H} (\mathbb{E}_{\mathcal{D}_T}[h(\mathbf{x})] - \mathbb{E}_{\mathcal{D}_S}[h(\mathbf{x})]) \right|. \qquad (23)$$

Note that due to the symmetric nature of the function space, we can just pick either side to lead with and drop the absolute value, yielding

$$|\gamma_T(h, f_T) - \gamma_S(h, f_T)| \leq \max_{h : \mathcal{X} \to [0,1], ||h||_L \leq \lambda + \lambda_H} \mathbb{E}_{\mathcal{D}_S}[h(\mathbf{x})] - \mathbb{E}_{\mathcal{D}_T}[h(\mathbf{x})]. \qquad (24)$$

$$\leq \alpha_{\lambda + \lambda_H}(\mathcal{D}_S, \mathcal{D}_T). \qquad (25)$$

$\square$

Compared to the bound with $\mathcal{H}$-divergence, $\alpha_{\lambda + \lambda_H}(\mathcal{D}_S, \mathcal{D}_T)$ gives a distance between source and target domain with continuous probabilistic estimates, and encourages the analysis of smoothness properties of the features.

Following the Theorem 2 in [4], we can also easily bound the target error $\gamma_T(f, g_T)$ by

$$\gamma_T(f, g_T) \leqslant \gamma_S(f, g_S) + \alpha_{\lambda + \lambda_{h*}}(\mathcal{D}_S, \mathcal{D}_T) + \gamma^*,$$

where $\gamma^* = \min_{f \in \mathcal{H}} \gamma_S(f, g_S) + \gamma_T(f, g_T)$ is the minimum error can be reached.

There is a further relationship to accuracy from this analysis. Assuming the accuracy is defined by taking the maximal class from the prediction, the accuracy of the classifier would be given by

$$e_S(f, g_S) = \mathbb{E}_{\mathcal{D}_S}[f(\mathbf{x}) 1_{g_S(\mathbf{x}) \geqslant .5} + (1 - f(\mathbf{x})) 1_{g_S(\mathbf{x}) < .5}]. \qquad (26)$$

## A.3   Bound on Weighted Domains

In this section, we will give the proof of our main Theorem 3.3. First, we define the optimal discrepancy $\gamma^*$.

$$
\begin{aligned}
\gamma^* &= \min_f \left[ \gamma_T(f, g_T) + \sum_{s=1}^S w_s \gamma_s(f, g_s) \right] \\
&= \gamma_T(f^*, g_T) + \sum_s^S w_s \gamma_s(f^*, g_s). \qquad (27)
\end{aligned}
$$

*Proof of Theorem 3.3.*

$$\gamma_T(f, g_T) \leq \gamma_T(f^*, g_T) + \gamma_T(f, f^*) \quad \text{(Triangle Inequality)}$$

$$= \gamma_T(f^*, g_T) + \gamma_T(f, f^*) - \sum_{s=1}^{S} w_s \gamma_s(f, f^*) + \sum_{s=1}^{S} w_s \gamma_s(f, f^*)$$

$$\leq \gamma_T(f^*, g_T) + \left| \gamma_T(f, f^*) - \sum_{s=1}^{S} w_s \gamma_s(f, f^*) \right| + \sum_{s=1}^{S} w_s \gamma_s(f, f^*)$$

$$\leq \gamma_T(f^*, g_T) + \alpha_{\lambda+\lambda^*} (\sum_{s=1}^{S} w_s \mathcal{D}_s, \mathcal{D}_T) + \sum_{s=1}^{S} w_s \gamma_s(f^*, g_s) + \sum_{s=1}^{S} w_s \gamma_s(f, g_s)$$

$$\leq \sum_{s=1}^{S} w_s \gamma_s(f, g_s) + \alpha_{\lambda+\lambda^*} (\sum_{s=1}^{S} w_s \mathcal{D}_s, \mathcal{D}_T) + \gamma^*$$

$\square$

In order to lower the target error, the discrepancy, or the $\alpha$-quantity needs to be small. This makes sense by saying that when source and target are uncorrelated to each other, the domain adaptation cannot succeed. Another interesting finding is that when $h$ is in a complex space, i.e. the classifier is strong, the discrepancy term $\gamma^*$ can be lowered down. However, the $\alpha$-quantity term will be large due to the uncertainty in the space. On the contrary, a restricted hypothesis $f$ will result in a large $\gamma^*$ and a small $\alpha$-quantity. This trade-off tells us that the classifier should remain in a reasonable complexity in order to make the target error small.

## B   Result on Text Data

For text domain adaptation task, we use the Amazon reviews dataset to perform sentiment analysis. This dataset contains reviews on product from four categories: books, electronics, DVD and kitchen. Each category forms a domain. Follows the preprocessing step of [14, 6], we choose the 5000 dimensional feature vector of unigrams and bigrams. These features are further encoded with mSDA [6]. Each item has a review ranking from one to five starts. We treat those with one to three stars as negative and four to five starts as positive. For each experiment, we leave one category out as target domain, while leaving the other three as source domains.

|  | Books | DVD | Electronics | Kitchen |
|---|---|---|---|---|
| Baseline | 79.52 | 81.54 | 83.49 | 85.78 |
| TCA [34] | 79.64 | 79.75 | 72.49 | 84.81 |
| SA [13] | 79.04 | 81.96 | 83.37 | 85.55 |
| ITL [36] | 79.60 | 81.90 | 82.75 | 85.25 |
| DANN [14] | 79.60 | 80.51 | 84.12 | 85.84 |
| MDANs [45] | 80.76 | **82.74** | 84.54 | 86.16 |
| MDMN | **81.21** | 82.37 | **84.63** | **86.56** |

Table 3: Accuracy on text sentiment classification.

The feature extractor is a neural network with three fully connected layers with 1000, 500 and 100 hidden units respectively. ReLU is used between layers. Both the label and domain predictors contain two fully connected layers with 100 hidden units for the first layer and the second layer gives the output. For the baseline model, it consists the feature extractor and the label predictor. For the TCA, SA and ITL methods, we use the extracted 100 dimensional output from feature extractor from the baseline model as input feature. The classification accuracy on each setting is listed in Table 3.

In this application, MDMN shows marginal improvement. Because this corpus is fairly large and the domains match well, there is less of an advantage to multiple domain adaptation. Regardless, the proposed approach performs well.

## C   Experiment Plots

### C.1   Digit Dataset visualization and Larger Plots on Digit Classification Result

In our digit number classification experiment in section 5.1, we use MNIST, MNISTM, USPS and SVHN dataset. MNIST and USPS are both black and white hand written digits. MNISTM is colored

MNIST dataset with a background added [14]. SVHN contains colored street number. A sample is given in figure C.1.

Figure 6: Sample of the digit datasets.

In Figure 7, we provide a larger version of the result in Section 5.1 to enhance clarity.

## C.2   EEG Subject-wise Result

We also include the subject-wise classification performance without subtracting the baseline. The baseline the from a recently proposed neural network based algorithms for EEG/LFP signal [23].

(a) Baseline

(b) DANN

(c) MDANs

(d) MDMN

Figure 7: Larger visualization of the digit classification task shown in Figure 4. Introduction to this experiment and analysis is given in Section 5.1.

(a) Accuracy on SEED dataset

(b) Accuracy on ASD dataset

Figure 8: Subject-wise performance on EEG datasets.