[Reviews · NeurIPS 2018]

Reviewer 1



This paper presents a technique for adapting from multiple sources to a target. The main goal of this paper is to mitigate negative transfer from unrelated domains. The authors use the now popular adversarial neural networks-style framework where the feature layer is trained to minimize the label loss + a loss function to measure how close the domains are in the feature space. The main contribution of the paper is to use a Wasserstein like distance between the source distributions and the target distribution for the second part. The authors further add a weighted importance to each of the source domains to represent the relevance of that domain. The weights for the domains are the output of the softmax over the Wasserstein distances of target domain to each of source domains. The authors also show an error bound over classifier discrepancies using a Wasserstein-like distance measure. The key idea about this bound is that it bounds the error on target domain by a weighted sum of errors on source domains and not with the max -- this implies that reducing the weight over a potentially unrelated source domain can mitigate its impact. Pros: 1. Well written and straightforward to understand. 2. Theorem 3.3 is a nice extension to previous results on multi domain adaptation. Cons: 1. Lack of experimental analyses. E.g. it needs ablation analysis to disentangle the contributions of Wasserstein distribution matching and weighted source matching. Suggestions: 1. Authors should motivate the results in section 3 in a better way in light of the proposed setup. It is not clear how theorem 3.3 relates to the proposed technique. 2. Section 2 will be much clearer if the authors put the algorithm in the main paper instead of the appendix. E.g. there is a circular dependency between source weights w_s and the parameters \theta_D and it is not entirely clear that this is handled iteratively until one looks at the algorithm. Read author response and satisfied.

Reviewer 2



This paper addresses the problem of multi-domain adaptation. We are faced with several (closely) related tasks, each one with limited resources to train a classifier. The classical approach is to map from one or several sources domains to the target domain, disregarding potential relations among the source domains. In this paper, it is proposed to also consider weighted relations between the source domains. The approach is named Multiple Domain Matching Network (MDMN). The paper is well written. I'm not an expert of transfer learning, but I was able to follow the motivation and main ideas of this work. The algorithm is evaluated for an image classification task and two tasks of the medical domain. Each time, N-1 tasks are considered as source domain and the Nth task as target. In image classification, the tasks are MNIST, MNISTM, USPS and SVHN. The proposed method nicely outperforms other domain adaption methods. I'm not working in computer vision, but it seems to me that the state-of-the-art performance on MNIST is better than 5.4% error. The goal of the medical tasks is to predict emotion or treatment stage given EEG measurements while subjects are watching dedicated movie clips. Again, MDNN outperforms published work. What is your recommendation when your approach should work best, in comparison to other domain adaptation frameworks ? When we have several source domains which are rather heterogeneous ? Does your algorithm scale with the number of source domains ? I guess that it does not improve on other domain adaptation or transfer learning approaches if you have only one source domain. I also wonder if your approach could be used to outperform the best known separate baselines of several tasks ? For instance, the current state-of-the-art on MNIST, ImageNet and FlickR ?

Reviewer 3



Title: Extracting Relationships by Multi-Domain Matching Summary Assuming that a corpus is compiled from many sources belonging to different to domains, of which only a strict subset of domains is suitable to learn how to do prediction in a target domain, this paper proposes a novel approach (called Multiple Domain Matching Network (MDMN)) that aims at learning which domains share strong statistical relationships, and which source domains are best at supporting to learn the target domain prediction tasks. While many approaches to multiple-domain adaptation aim to match the feature-space distribution of *every* source domain to that of the target space, this paper suggests to not only map the distribution between sources and target, but also *within* source domains. The latter allows for identifying subsets of source domains that share a strong statistical relationship. Strengths Paper provides a theoretical analysis that yields a tighter bound on the weighted multi-source discrepancy. Approach yields state-of-the-art performance on image, text and multi-channel time series classification tasks. Weaknesses Tighter bound on multi-source discrepancy depends on the assumption that source domains that are less relevant for the target domain have lower weights. While intuitively, this may seem obvious, there is no guarantee that in practice, the irrelevant source domains can reliably be identified. No commitment that source code may get released. Questions L 96: Is it intended that the sum runs over all domains but s, including the target domain S? L120: Why is the Wasserstein distance not affected by a varying feature scale in practice? There is a shift in notation in Section 3 where the target domain is not denoted by T while the source domains are denoted by s=1...S. In Section 2, the source domains were defined as s=1,...,S-1 while the single target domain was defined as S. Theorem 3.3 shows that weighting yields a tighter bound given that irrelevant domains are assigned small weights. However, what happens if the algorithm fails to assign small weights to irrelevant domains or, in the most adverse case, if the least relevant domains get assigned the highest weights? More general: for which weight distributions does Theorem 3.3 provide tighter bounds? To what number of source domains does the provided method scale? A total of 21 domains may still be a small number if this method were to be applied to other tasks. What potential does MDMN have to be extended or applied to fuzzy domains, i.e., where the source data set does not induce a canonical partitioning into domains? Comments Editorial Notes L 36 is be -> is L 219 develop -> development L 287 An -> A L 320 [add reference] L 321 domains is large L 342 state-of-the0art -> state-of-the-art --------------- Thanks to the authors for their detailed responses and addressing my questions. I am looking forward to the release of the code.